# Identifying Policy Gaps in a COVID-19 Online Tool Using the Five-Factor Framework

**Janet Michel [1,*], David Evans [2], Marcel Tanner [3,4] and Thomas C. Sauter [1]**

[1] Department of Emergency Medicine, Inselspital, University Hospital, University of Bern, 3010 Bern, Switzerland

[2] World Bank Health Economist, 7 Bis Avenue de la Paix, 1202 Geneva, Switzerland

[3] Department of Epidemiology and Public Health, Swiss Tropical and Public Health Institute, 4002 Basel, Switzerland

[4] Faculty of Science, University of Basel, 4001 Basel, Switzerland

[*] Correspondence: janetmichel71@gmail.com

**Abstract:** Introduction: Worldwide health systems are being faced with unprecedented COVID-19-related challenges, ranging from the problems of a novel condition and a shortage of personal protective equipment to frequently changing medical guidelines. Many institutions were forced to innovate and many hospitals, as well as telehealth providers, set up online forward triage tools (OFTTs). Using an OFTT before visiting the emergency department or a doctor's practice became common practice. A policy can be defined as what an institution or government chooses to do or not to do. An OFTT, in this case, has become both a policy and a practice. Methods: The study was part of a broader multiphase sequential explanatory design. First, an online survey was carried out using a questionnaire to $n = 176$ patients who consented during OFTT usage. Descriptive analysis was carried out to identify who used the tool, for what purpose, and if the participant followed the recommendations. The quantitative results shaped the interview guide's development. Second, in-depth interviews were held with a purposeful sample of $n = 19$, selected from the OFTT users who had consented to a further qualitative study. The qualitative findings were meant to explain the quantitative results. Third, in-depth interviews were held with healthcare providers and authorities ($n = 5$) that were privy to the tool. Framework analysis was adopted using the five-factor framework as a lens with which to analyze the qualitative data only. Results: The five-factor framework proved useful in identifying gaps that affected the utility of the COVID-19 OFTT. The identified gaps could fit and be represented by five factors: primary, secondary, tertiary, and extraneous factors, along with a lack of systems thinking. Conclusion: A theory or framework provides a road map to systematically identify those factors affecting policy implementation. Knowing how and why policy practice gaps come about in a COVID-19 OFFT context facilitates better future OFTTs. The framework in this study, although developed in a universal health coverage (UHC) context in South Africa, proved useful in a telehealth context in Switzerland, in Europe. The importance of systems thinking in developing digital tools cannot be overemphasized.

**Keywords:** utility; five-factor framework; policy gaps; COVID-19 OFFT; systems thinking

## 1. Introduction

Worldwide health systems are being faced with unprecedented COVID-19-related challenges ranging from the problems of a novel condition and a shortage of personal protective equipment to frequently changing medical guidelines [1]. Online forward triage tools (OFTTs) facilitate the interaction between a user/human and a computer system and gives a recommendation on what to do based on the input received [1–3]. Many institutions were forced to innovate and many hospitals as well as telehealth providers set up online forward triage tools (OFTTs) [1]. Using an OFTT before visiting the emergency department or a doctor's practice, therefore, became common practice.

A policy is defined as what an institution or government chooses to do or not to do [4]. The policy process is widely conceptualized as six stages: (1) problem emergence; (2) agenda setting; (3) consideration of policy options; (4) decision-making; (5) implementation; (6) evaluation [5]. The policy cycle is also described as policy development, policy communication, policy implementation, and policy evaluation [6]. In light of the above, the COVID-19 OFTT is both a policy and a practice.

The SARS-CoV-2 pandemic accelerated the adoption of telehealth services, particularly OFTTs [1,7,8]. OFTTs have been reported to reduce the health system burden, to inform and direct patients toward the appropriate level of care, e.g., to test or not to test, how to conduct self-care, as well as how to relieve anxiety [1,9]. The Inselspital, University Hospital Bern, set up an OFTT, coronatest.ch, on 2 March 2020, to cope with the influx of SARS-CoV-2 patients. The tool was updated regularly, based on the changing testing criteria issued by the Swiss Federal Office of Public Health [1].

Due to the urgency of the situation, patients or potential tool users could not be consulted during tool development but the tool was pilot-tested by ER physicians. With no active advertisement, the tool was offered by the Inselspital hospital. It is noteworthy that involving end users, policy implementers, and beneficiaries facilitates successful policy implementation [10–12].

Evaluating policy implementation facilitates learning which in turn leads to success and positive outcomes. Many OFTTs have however not been evaluated. Identifying frameworks that work is the first step in that direction. A theory or framework provides a road map for systematic evaluation, identifying factors that actors perceive as affecting implementation. We utilized the five-factor framework as our analytic tool [11]. The purpose of this manuscript is to assess the utility of the five-factor framework in identifying how and why policy–practice gaps come about in a COVID-19 OFTT implementation context.

## 2. Methods

### 2.1. Context

The Emergency Department, Inselspital University Hospital Bern decided to set up coronatest.ch, an OFTT, in March 2020. The assessment tool, coronatest.ch, was designed to deal with an influx of patients during the SARS-CoV-2 pandemic.

### 2.2. Study Aim

The aim of the study was to assess the utility of the five-factor framework in identifying how and why policy–practice gaps came about within the context of a COVID-19 online forward triage tool.

### 2.3. Study Design

This study was part of a broader multiphase sequential explanatory study.

Participants included OFTT users aged 18 and above who used the Insel COVID-19 OFTT between 2 March and 12 May 2020. A total of 6272 users consulted the COVID-19 OFTT and quantitative data was collected from 560 participants, who consented to a follow-up survey and provided valid email addresses. A total of $n = 176$ out of the 560 participants completed the online survey. First, a descriptive analysis was carried out to identify who used the online tool and for what purpose, and if, indeed, they followed the recommendations (see Table 1, below). The quantitative results guided the interview guide development. Second, in-depth interviews were held with a purposeful sample, $n = 19$, selected from the OFTT users who had consented to a further qualitative study. The qualitative findings were meant to explain the quantitative results. Third, in-depth interviews were held with healthcare providers and authorities ($n = 5$) who were privy to the analytical tool due to their professional roles. Framework analysis was adopted, using the five-factor framework as a lens, to analyze the qualitative data only.

**Table 1.** The socio-demographic characteristics of the survey participants (quantitative data) [13].

|  | Total | (*n* = 176) | Female | (*n* = 101) | Male | (*n* = 75) | *p*-Value * |
|---|---|---|---|---|---|---|---|
| **Age [mean, SD]** | 50.1 | [±15.4] | 45.9 | [±14.1] | 55.7 | [±15.4] | <0.001 |
| **Education** |  |  |  |  |  |  |  |
| Did not want to answer | 6 | [3.4] | 3 | [3.0] | 3 | [4.0] |  |
| University | 120 | [68.2] | 67 | [66.3] | 53 | [70.7] |  |
| Higher secondary school | 27 | [15.3] | 17 | [16.8] | 10 | [13.3] |  |
| Lower secondary school | 23 | [13.1] | 14 | [13.9] | 9 | [12.0] | 0.871 |
| **Income per month** |  |  |  |  |  |  |  |
| Did not want to answer | 29 | [16.5] | 17 | [16.8] | 12 | [16.0] |  |
| <4000 CHF | 26 | [14.8] | 20 | [19.8] | 6 | [8.0] |  |
| 4000–6000 | 42 | [23.9] | 27 | [26.7] | 15 | [20.0] |  |
| >6000 | 79 | [44.9] | 37 | [36.6] | 42 | [56.0] | 0.037 |
| **Work** |  |  |  |  |  |  |  |
| Did not want to answer | 33 | [18.8] | 14 | [13.9] | 19 | [25.3] |  |
| Employed | 106 | [60.2] | 64 | [63.4] | 42 | [56.0] |  |
| Self-employed | 24 | [13.6] | 13 | [12.9] | 11 | [14.7] |  |
| Unemployed | 3 | [1.7] | 3 | [3.0] | 0 | [0.0] |  |
| Lost work (COVID-19) | 1 | [0.6] | 1 | [1.0] | 0 | [0.0] |  |
| Student/trainee | 9 | [5.1] | 6 | [5.9] | 3 | [4.0] | 0.236 |
| **Insurance** |  |  |  |  |  |  |  |
| Do not know | 5 | [2.8] | 3 | [3.0] | 2 | [2.7] |  |
| General | 68 | [38.6] | 39 | [38.6] | 29 | [38.7] |  |
| Telemedicine | 12 | [6.8] | 6 | [5.9] | 6 | [8.0] |  |
| GP | 83 | [47.2] | 47 | [46.5] | 36 | [48.0] |  |
| Other | 8 | [4.5] | 6 | [5.9] | 2 | [2.7] | 0.859 |
| **Nationality** |  |  |  |  |  |  |  |
| Did not want to answer | 1 | [0.6] | 1 | [1.0] | 0 | [0.0] |  |
| Switzerland | 147 | [83.5] | 80 | [79.2] | 67 | [89.3] |  |
| Germany | 13 | [7.4] | 8 | [7.9] | 5 | [6.7] |  |
| French | 1 | [0.6] | 0 | [0.0] | 1 | [1.3] |  |
| Italy | 3 | [1.7] | 2 | [2.0] | 1 | [1.3] |  |
| Other Europe | 4 | [2.3] | 3 | [3.0] | 1 | [1.3] |  |
| Other | 7 | [4.0] | 7 | [6.9] | 0 | [0.0] | 0.202 |

* Chi-squared for categorical variables and Wilcoxon rank sum test for continuous variables; data are total number and percentage if not mentioned otherwise

## 3. Qualitative Data Collection

The qualitative interviews were conducted with purposefully selected key informants who gave their consent during the survey (see Table 2 below).

**Table 2.** Key informants (patients, healthcare providers, and authorities).

| Key Informants | Male | Female | Total |
|---|---|---|---|
| OFTT users—patients | 10 | 9 | 19 |
| Healthcare providers and authorities | 1 | 4 | 5 |
| **Total** | **11** | **13** | **24** |

Video rather than face-to-face interviews were held with most participants in September 2020, due to social-distancing rules. A combination of video and telephonic interviews was conducted with three participants who encountered technical difficulties and a telephone-only interview was held with one lady, aged over 65, who had no computer access. Three face-to-face interviews were held with three key informants: one was a hospital healthcare worker and two other key informants worked close to Bern University Hospital. A semi-structured interview guide, informed by the quantitative results, was used (see Supplementary Materials Figures S1–S3). This guide was adapted iteratively throughout the data collection period. Two qualitative researchers sat in each session,

fielding questions in turn. All interviews were conducted in German by two researchers who were fluent in both English and German. The interviews lasted between 45 min to one and a half hours. Two audio recorders were used in each session. All participants gave individual written consent as well as oral consent to their being recorded at the beginning of each session (see Table 2 for a summary of the key informants).

*Qualitative Data Analysis*

All audio recordings were transcribed verbatim, analyzed, and triangulated with the results from the quantitative data. Qualitative narratives were explored for their fit with the five factors of the analytic framework [11]. Two qualitative researchers analyzed the transcripts independently and developed and agreed on a code book. All the concepts fitted into the five factor themes.

## 4. Measures to Ensure the Trustworthiness of the Data

To ensure dependability, the data collection process and analysis were performed iteratively, continuously adjusting our interview guide to capture newly emerging themes. Two qualitative researchers kept reflexive journals and debriefed at the end of each interview throughout the data collection phase. A comprehensive description of the participants, context, and data collection process has been outlined here to ensure transferability. Data were managed and analyzed with the aid of MAXQDA2020.

### 4.1. Ethics Approval

Our study is embedded in an online forward triage tool set up by the Insel University Hospital within a pandemic setting, primarily to prevent health-system overload. The evaluation of the usefulness of this tool to the health stakeholders, patients, healthcare providers, and health authorities was deemed to be a quality evaluation; hence, the ethics committee of the province (canton) of Bern, Switzerland, waived the need for a full ethical review (Req-2020-00289) on the 23 March 2020 and granted us permission to carry out the study.

### 4.2. Central Questions

How well do the identified themes fit into the five-factor framework?

How well does the five-factor framework explain why and how the policy–practice gaps came about?

### 4.3. The Five-Factor Framework [11]

A theory or framework provides a road map for systematically identifying those factors perceived by all stakeholders as affecting implementation. With the aid of the five-factor framework, we identified COVID-19 OFTT (coronatest.ch) policy gaps. This framework, developed in a universal health coverage (UHC) context [11], goes beyond identifying barriers and facilitators of policy to explain how and why these policy–practice gaps came about.

### 4.4. Five Groups of Factors Identified as Bringing about Policy–Practice Gaps

(1) Primary factors stem from a direct lack of a critical component for policy implementation, whether tangible or intangible—resources, the policy itself, information, motivation, power, and context;

(2) Secondary factors stem from a lack of efficient processes or systems, e.g., budget processes, financial delegations, communication channels, top-down directives, supply chains, supervision, and performance management processes;

(3) Tertiary factors stem from human factors—perception, cognition, and calculated human responses to a lack of primary, secondary, and or extraneous factors as coping mechanisms (ideal reporting and audit-driven compliance);

(4) Extraneous factors stem from beyond the health system—economy, weather, climate, and drought;

(5) An overall lack of systems thinking also brings about this type of gap. See Figure 1 below.

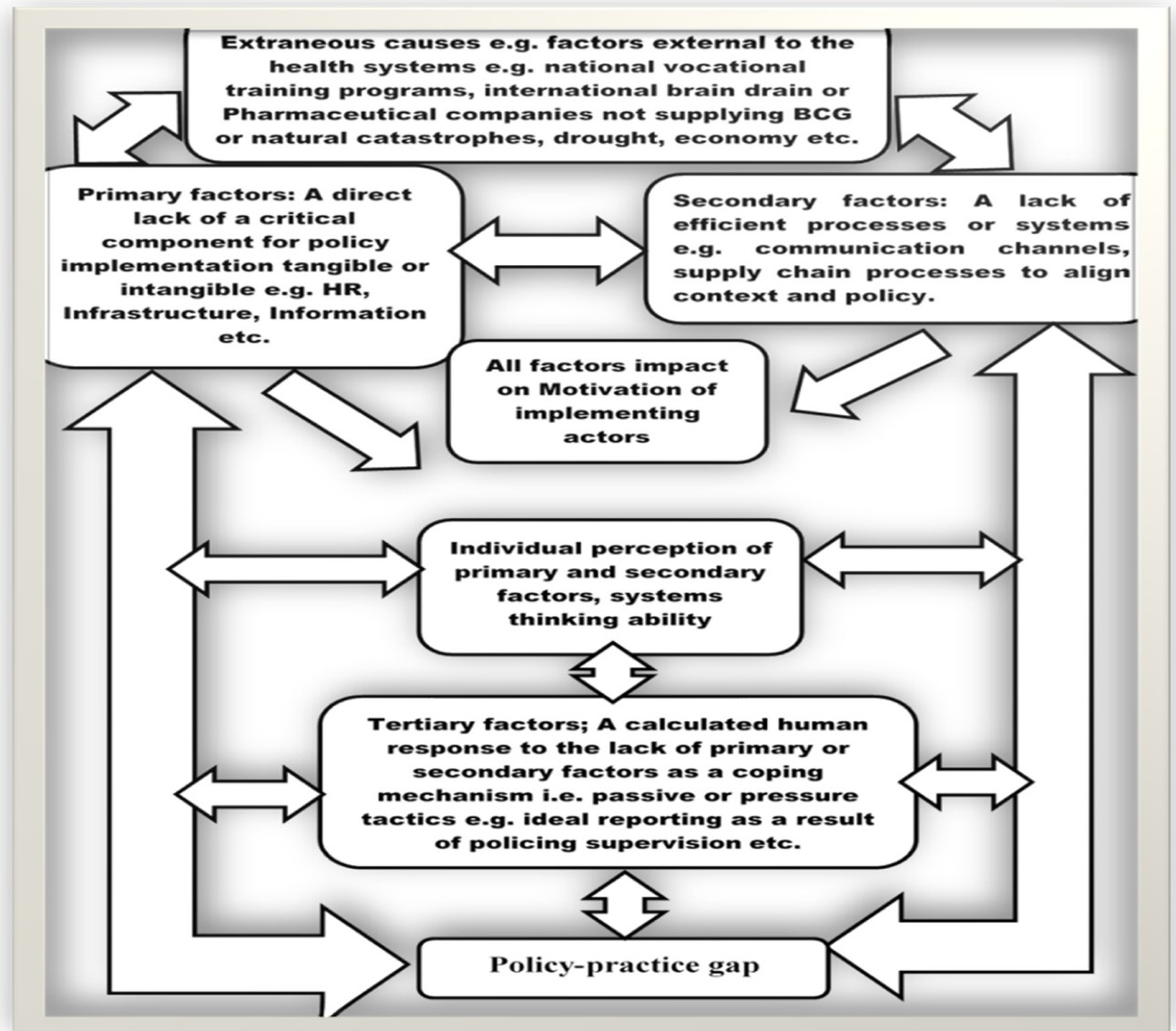

**Figure 1.** The five-factor framework [11].

## 5. Findings

*5.1. Primary Factors Stemming from the Direct Lack of a Critical Component for Policy Implementation, Whether Tangible or Intangible—Resources, Information, Motivation, and Power*

The policy itself, regarding the use of OFTT to reduce the health-system burden, was shown to be good in itself. Most of the participants, however, discovered the tool by chance, as the tool itself was not advertised. There was no coordinated way of communicating the tool's availability to other healthcare providers either.

> *"The tool is meant for adults. A similar tool that is child-specific would be very helpful."*

> Key informant 2 (healthcare provider)

The first OFTT, coronatest.ch, was adult-oriented and so child-specific information was missing. The first interviews revealed this lack, which led to the birth of another initiative, the launch of coronabambini, a child-specific OFTT [14].

*5.2. Secondary Factors Stemming from a Lack of Efficient Processes or Systems—Budget Processes, Limited Financial Delegation, Top-Down Directives, Communication Channels, Supply-Chain Processes, Ineffective Supervision, and Performance Management Systems*

The availability of the OFTTs was not communicated widely; neither were they advertised. Notwithstanding the communication challenge, many participants reported using the tool and receiving the recommendation to be tested, only to be met with test-kit shortages. Others reported that their GPs and pediatricians were not aware of the tool and so refused to give them the test. Other healthcare providers reported that shortages in terms of test availability prevented them from doing so.

> *"We did not have sufficient test kits at the beginning; we ran out and could not test."*
>
> Key informant (healthcare provider)

*5.3. Tertiary Factors Stemming from Human Factors—Perception and Cognition, and the Calculated Human Responses to a Lack of Primary, Secondary, and/or Extraneous Factors as Coping Mechanisms (Ideal Reporting and Audit-Driven Compliance with Core Standards)*

The system is only as good as the people within the system. The GPs responded in different ways when patients suspected that they had COVID-19, as revealed below:

> *"When I asked for a test, my GP told me that this is* [a] *hysterical* [response], *everyone now thinks that they have COVID-19."*
>
> Key informant (patient)

> *"What is interesting is that the GPs were open to testing children, while the pediatricians refused* [to test] *the children."*
>
> Key informant (patient)

*5.4. Extraneous Factors Stemming from Beyond the Health System (National Vocational Training, Leading to a National Shortage of Plumbers)*

COVID-19, a novel infection, took the world by surprise. There was a lack of knowledge of the disease signs and symptoms, progression, and even management. This made the guidelines change frequently as a result, with sometimes conflicting information being given, including those concerning mask mandates.

> *"The whole pandemic took us all by surprise."*
>
> Key informant (health authority)

*5.5. An Overall Lack of Systems Thinking*

An OFTT is dependent upon other parts of the system, for example, the supply chain, testing centers, and the readiness of the patients to follow recommendations. Fear, social media, rumors, and disinformation, although not primarily health system factors, also affected attitudes to OFTT testing recommendations. Some participants revealed the following:

> *"Many people did not test for fear of a positive test result. They would rather not know."*
>
> Key informant (patient)

## 6. Discussion

We assessed the utility of the five-factor framework in identifying how and why policy–practice gaps came about within a COVID-19 online forward triage tool. The themes that emerged from the qualitative data could fit into the five factors: primary, secondary, tertiary, and extraneous factors, along with a lack of systems thinking, and helped explain how and why policy–practice gaps come about in the context of a COVID-19 OFTT. See Table 3 and Figure 1 above.

**Table 3.** Summary of the emergent themes.

| Theme | Category | Unit Meaning |
|---|---|---|
| **1. Primary factors stemming from a direct lack of a critical component for policy implementation, tangible or intangible—resources, information, motivation, power** | Policy communication | - Often, it was not advertised |
| **2. Secondary factors stemming from a lack of efficient processes or systems—budget processes, limited financial delegations, top-down directives, communication channels, supply chain processes, ineffective supervision, and performance management systems** | Supply chain challenges Infrastructural challenges | - Test kit shortages<br>- Laboratory testing capacity |
| **3. Tertiary factors stemming from human factors—perception and cognition and calculated human responses to a lack of primary, secondary, and or extraneous factors, as coping mechanisms (ideal reporting and audit-driven compliance with core standards)** | Human factors | - GPs told patients that they are being hysterical, they cannot have COVID-19 |
| **4. Extraneous factors stemming from beyond the health system (national vocational training, leading to a national shortage of plumbers)** | Factors beyond the health system | - A novel condition; therefore, no one knew what to expect with COVID-19<br>- Economic factors |
| **5. An overall lack of systems thinking** | The utility of the tool in testing is affected by so many factors | - Test kit shortages, psychological readiness to test, the healthcare provider's trust in the tool |

*6.1. Primary Factors Stemming from the Direct Lack of a Critical Component for Policy Implementation, Whether Tangible or Intangible—Resources, Information, Motivation, and Power*

The Inselspital Emergency Department responded to the high volumes of calls by setting up an OFTT, so as to reduce the burden on the health system. Neither the healthcare providers nor the patients were involved in tool development. The tool itself was not advertised and those that used the tool discovered it by chance, revealing an information gap. This five-factor framework recommends the involvement of both policymakers and policy implementers and their beneficiaries, where appropriate, as a way of achieving the buy-in and uptake of future tools. This shortcoming is also highlighted by Greenhalgh et al. [15]. The communication of this policy, although closely related to policy development, is, in itself, very important for successful policy implementation [15]. This identified gap might have resulted in a low buy-in from healthcare providers and in some patient groups who are not technology-aware being excluded.

*6.2. Secondary Factors Stemming from a Lack of Efficient Processes or Systems-Budget Processes, Limited Financial Delegations, Top-Down Directives, Communication Channels, Supply Chain Processes, Ineffective Supervision, and Performance Management Systems*

The five-factor framework revealed that healthcare providers were not involved, and patients reported being refused a test by some doctors who were not aware of the tool

and, hence, did not trust the recommendations affecting policy implementation (OFFT). Shortages of test kits were reported at the beginning of the pandemic. This points toward supply-chain issues, a secondary factor in the five-factor framework. The shortage of test kits represents a supply-chain issue that affected the usefulness of the OFTT on testing, since those who needed a test could not access one. In line with our findings, supply-chain issues can impact policy implementation, either positively or negatively [16].

### 6.3. Tertiary Factors Stemming from Human Factors—Perception and Cognition and Calculated Human Responses to a Lack of Primary, Secondary, and or Extraneous Factors as Coping Mechanisms (Ideal Reporting and Audit-Driven Compliance with Core Standards)

The utility of the assessment tool in reducing the health-system burden was acknowledged by patients, providers, and authorities. Some aspects, such as the utility of the tool in relieving fear and anxiety, were acknowledged by patients but were disputed by healthcare providers. Human factors play a role in implementation; being a patient or healthcare provider changes how one perceives the utility of a tool [11,17]. The importance of human factors in policy implementation cannot be overemphasized [18].

### 6.4. Extraneous Factors Stemming from beyond the Health System (National Vocational Training, Leading to a National Shortage of Plumbers)

COVID-19 is a novel condition. Neither the authorities nor the clinicians had knowledge of its pathology at the beginning of the outbreak, leading to ever-changing guidelines and conflicting messages since there was no prior knowledge to fall back on. This underlies the fact that some issues affecting implementation go far beyond the health system. In addition, a number of OFTT users who received the recommendation to be tested for COVID-19 did not go on to do so. Many cited the fear of losing income and possibly their jobs, revealing how factors beyond the health system, such as economic factors, affected OFTT implementation, concurring with the findings reported elsewhere [18]. Contrary to our findings, OFTTs have been associated with risk aversion, resulting in increased healthcare service use rather than the reduction of the healthcare system burden [19,20]. It is worth highlighting that while as OFTTs can educate clients and provide information on symptoms, they cannot talk to the patient, touch, feel, or look the patient in the eye, a vital shortcoming that underlies the importance of the human factor in health care [20].

### 6.5. An Overall Lack of Systems Thinking

A proportion of people who received a recommendation to test did not do so. This finding was associated with the psychological readiness of patients to test, which, in turn, was influenced by the fear of receiving a positive test result. Even after resolving the supply-chain issues, having the test kits alone did not resolve this issue, highlighting the interconnectedness of things and the importance of systems thinking. Senge also emphasizes the importance of systems thinking in policy implementation [21].

### 6.6. Strengths and Limitations

1. Our study tested the utility of a five-factor framework, thereby contributing to the body of OFTT evaluation frameworks.
2. Knowing how and why policy practice gaps come about in a COVID-19 OFFT context facilitates success in future and better OFTTs.
3. Our study demonstrated the importance of systems thinking in developing digital tools and this importance cannot be overemphasized.
4. The key informants were sampled from online OFTT users. The perspectives of key informants that do not have access to or do not use OFTTs are not represented.

Few OFTTs have been evaluated; one of the major stumbling blocks is a lack of OFTT evaluation frameworks. Our study tested the utility of a five-factor framework, thereby contributing to the body of OFTT evaluation frameworks. Identifying frameworks that work is the first step. A theory or framework provides a road map for systematic evaluation,

identifying those factors that actors perceive as affecting implementation. Knowing how and why policy practice gaps come about in a COVID-19 OFFT context facilitates success in future and better OFTTs. The five-factor framework, although developed in a universal health coverage (UHC) context in South Africa, proved useful in identifying policy–practice gaps in a COVID-19 OFTT used in Switzerland, in Europe. Our study demonstrated the importance of systems thinking in developing digital tools and this cannot be overemphasized. The key informants in this study were sampled from online OFTT users. The perspectives of key informants that do not have access to or do not use OFTTs are, thus, not represented. To the best of our knowledge, this selection bias could not be prevented due to data protection regulations, which impose voluntary participation and prohibit the technically possible automatic tracking of participants.

## 7. Conclusions

The five-factor framework proved useful in identifying gaps that affected the utility of the COVID-19 OFTT. The identified gaps could fit and be represented by the five factors: primary, secondary, tertiary, and extraneous factors, along with a lack of systems thinking. The framework, although developed in a universal health coverage (UHC) context in South Africa, proved useful in a telehealth context in Switzerland, in Europe. A theory or framework provides a road map to systematically identify those factors affecting policy implementation [22]. Knowing how and why policy practice gaps came about in a COVID-19 OFFT context facilitates success in future and better OFTTs. These findings are encouraging, and we recommend that others should test this framework in other settings and contexts to assess its utility in identifying how and why policy–practice gaps come about. This is particularly important to address, as evidence repeatedly points out that policies are rarely translated into practice [11,23–26].

**Supplementary Materials:** The following supporting information can be downloaded at: https://www.mdpi.com/article/10.3390/systems10060257/s1, Figure S1: Interview Guide: Key Informants—Patients; Figure S2: Interview Guide: Key Informants—Healthcare Providers; Figure S3: Interview Guide: Key Informants—Health Authorities.

**Author Contributions:** Study design and ideas: J.M., D.E., M.T. and T.C.S.; qualitative data analysis, J.M., D.E., M.T. and T.C.S.; writing of the first draft: J.M., D.E., M.T. and T.C.S.; revision of final draft and approval, J.M., D.E., M.T. and T.C.S.; project administrator, T.C.S. All authors have read and agreed to the published version of the manuscript.

**Funding:** Swiss National Science Foundation (Project ID: 196615).

**Data Availability Statement:** Due to the nature of the study (OFTT), participants did not agree that their data should be shared publicly. The data that support the findings are available on reasonable request. Please contact the corresponding author, J.M.

**Conflicts of Interest:** The authors declare no conflict of interest. The present manuscript is partially funded by the Swiss National Foundation (Project ID: 196615). The funder has no influence on the content of the manuscript or the decision to publish it. T.C.S. holds the endowed professorship for emergency telemedicine at the University of Bern, Switzerland. The funder, Touring Club Switzerland, has no influence on the research performed, the content of any manuscript, or any decision to publish. All other authors have nothing to disclose.

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
