# Peer review of "Identifying Policy Gaps in a COVID-19 Online Tool Using the Five-Factor Framework"

_systems, doi:10.3390/systems10060257_

Round 1

Reviewer 1 Report

In general, the methodology in this research must be improved. It is necessary to clearly state the period of surveying the participants, as well as the selection of questions or topics that were covered by the research. Also, in this study the sampling strategy is not completely clear. Inclusion and exclusion criteria for the selection of respondents were not specified. It was stated that the research was part of a broader multiphase sequential explanatory study, but the details were not described. In the part of the research related to the findings, the characteristics of the patients who were interviewed were not represented. Therefore, it is not possible to determine the possible connection of their characteristics in the context with the answers they gave. The discussion also needs to be supplemented by a comparison with other studies that have examined the disadvantages of other tools that may have been used during the Covid-19 pandemic in other areas.

Author Response

                                                                                     Inselspital

                                                                                     Freiburgstrasse 18

                                                                                     3010 Bern

                                                                                     30 November 2022

Editor IJERPH MDPI

Dear Editor

RE: Identifying Policy Gaps in a COVID-19 online tool using the five-factor framework

We thank the reviewers for their invaluable comments on our manuscript; Identifying Policy Gaps in a COVID-19 online tool using the five-factor framework.” We have carefully taken all the reviewers’ comments into account and are pleased to submit a revised version herewith. We indicate below how we have changed the manuscript and respond to the reviewers’ comments item by item.

Sincerely,

Janet Michel, RN, BA Cur, MPH, PHD

David Evans, PhD

Marcel Tanner, Professor, PhD, Epidemiologist, MPH

Thomas C Sauter MD, MME, Professor

Reviewer 1

Comments and Suggestions for Authors: In general, the methodology in this research must be improved. It is necessary to clearly state the period of surveying the participants, as well as the selection of questions or topics that were covered by the research. Also, in this study the sampling strategy is not completely clear. Inclusion and exclusion criteria for the selection of respondents were not specified.  It was stated that the research was part of a broader multiphase sequential explanatory study, but the details were not described. In the part of the research related to the findings, the characteristics of the patients who were interviewed were not represented. Therefore, it is not possible to determine the possible connection of their characteristics in the context with the answers they gave.

*Author response

We thank the reviewers for this important question. We have now added the information on participant, sampling and characteristics. The section now reads as follows; Lines 112 -160

Study design

This was part of a broader multiphase sequential explanatory study.

Participants included OFTT users aged 18 and above that used the Insel COVID-19 OFTT between March 2nd and May 12th, 2020. A total of 6 272 users consulted the COVID-19 OFTT and quantitative data was collected from 560 participants that consented to a follow-up survey and provided valid email addresses. A total of n = 176 out of the 560 participants completed the online survey.  Descriptive analysis was carried out to identify who used the online tool and for what, and if indeed they followed the recommendations. The quantitative results guided the interview guide development. Second, in-depth interviews were held with a purposeful sample n=19 selected from the OFTT users that had consented to a further qualitative study. The qualitative findings were meant to explain the quantitative results. Third, in-depth interviews were held with health care providers and authorities (n =5) that were privy to the tool due to their professional roles.  Frame work analysis was adopted using the five-factor framework as a lens to analyse the qualitative data only.

Table 1 socio-demographic characteristics of survey participants (quantitative data) 14

Total

(n=176)

Female

(n=101)

Male

(n=75)

P-value*

Age [mean, SD]

50.1

[±15.4]

45.9

[±14.1]

55.7

[±15.4]

<0.001

Education

Not want to answer

6

[3.4]

3

[3.0]

3

[4.0]

University

120

[68.2]

67

[66.3]

53

[70.7]

Higher secondary school

27

[15.3]

17

[16.8]

10

[13.3]

Lower secondary school

23

[13.1]

14

[13.9]

9

[12.0]

0.871

Income per month

Not want to answer

29

[16.5]

17

[16.8]

12

[16.0]

<4000 CHF

26

[14.8]

20

[19.8]

6

[8.0]

4000 - 6000

42

[23.9]

27

[26.7]

15

[20.0]

>6000

79

[44.9]

37

[36.6]

42

[56.0]

0.037

Work

Not want to answer

33

[18.8]

14

[13.9]

19

[25.3]

Employed

106

[60.2]

64

[63.4]

42

[56.0]

Self-employed

24

[13.6]

13

[12.9]

11

[14.7]

Unemployed

3

[1.7]

3

[3.0]

0

[0.0]

Lost work (Covid-19)

1

[0.6]

1

[1.0]

0

[0.0]

Student/trainee

9

[5.1]

6

[5.9]

3

[4.0]

0.236

Insurance

Don't know

5

[2.8]

3

[3.0]

2

[2.7]

General

68

[38.6]

39

[38.6]

29

[38.7]

Telemedicine

12

[6.8]

6

[5.9]

6

[8.0]

GP

83

[47.2]

47

[46.5]

36

[48.0]

Other

8

[4.5]

6

[5.9]

2

[2.7]

0.859

Nationality

Not want to answer

1

[0.6]

1

[1.0]

0

[0.0]

Switzerland

147

[83.5]

80

[79.2]

67

[89.3]

Germany

13

[7.4]

8

[7.9]

5

[6.7]

French

1

[0.6]

0

[0.0]

1

[1.3]

Italy

3

[1.7]

2

[2.0]

1

[1.3]

Other Europe

4

[2.3]

3

[3.0]

1

[1.3]

Other

7

[4.0]

7

[6.9]

0

[0.0]

0.202

Qualitative data collection

The qualitative interviews were conducted with purposefully selected key informants who gave their consent during the survey. See table 2 below.

Video rather than face to face interviews were held with most participants in September 2020 due to social -distancing rules. A combination of video and telephonic interviews were conducted with three participants who had technical challenges and a telephone only interview was held with one lady, aged above 65, who had no computer access. Three face to face interviews were held with three key informants: one that was a hospital health care worker, and two key informants who worked close to Bern university hospital. A semi-structured interview guide informed by the quantitative results was used (see supplementary info). This guide was adapted iteratively throughout the data collection period. Two qualitative researchers sat in each session fielding questions in turns. All interviews were conducted in German by two researchers fluent in both English and German. The interviews lasted between 45 minutes to one and a half hours. Two audio-recorders were used in each session. All participants gave individual written consent as well as oral consent to the recording at the beginning of each session. See Table 1 and 2 for summary of Key Informants.

Table 2 Key informants (patients, health care providers and authorities)

Key Informants

Male

Female

Total

 OFTT user- patients

10

9

19

Health care -provider and authorities

1

4

5

Total

11

13

24

Reviewer comment: The discussion also needs to be supplemented by a comparison with other studies that have examined the disadvantages of other tools that may have been used during the Covid-19 pandemic in other areas.

*Author response

We thank the reviewers for this comment. We have now included this info. See lines 300-307

Many cited the fear of losing income and possibly their jobs, revealing how factors beyond the health system, economic factors affected OFTT implementation concurring with findings elsewhere.19 Contrary to our findings,  OFTTs have been associated with risk aversion, resulting in an increased health care service use rather than the reduction of health care system burden. 1.9,20It worth highlighting that while as OFTTs can educate clients, provide information on symptoms, they cannot talk to the patient, touch, feel or look the patient in the eyes, a vital shortcoming that underlies the importance of the human factor in health care21

Reviewer 2 Report

Thank you for the opportunity to review Identifying Policy Gaps in a COVID-19 online tool using the five-factor framework. I understand the paper is trying to test the utility of using a five-factor framework to understand policy-practice gaps in the implementation of an online forward triage tool. The framework is based on a Michel et al 2019 paper in Journal of Global Health Reports.

I am not a health practitioner responsible or implanting policy and practice, but a qualitative digital health researcher and use this lens review this paper. As such I have some major and minor comments that would improve the paper and make it more applicable to a wider readership.

Introduction:

Can you please define an OFTT from the patient perspective? What do patients do with and OFTT? It is not clear to me what the OFTT in this paper is doing or what OFTTs are in general. Please explain clearly what your OFTT is and provide 2 others contrasting OFTT examples from other hospitals or health services.

The policy sentence in the introduction (line 69) is completely jarring. You mention OFTTs; but them immediately change topics to move into policy. Both could be better explained, and you need to draw a parallel between these. It is not clear to me if you think is OFTT is a policy. If that is the case, please say it before introducing a new paragraph on policy. If an OFTT is a practice, then define it as per your recommendation that you are trying to explore policy-practice gaps. If it is both, you need to define how your OFTT is both, provide the policy and practice guidelines from your organisation.

It is not clear if you developed or simply utilised a 5-factor framework for the evaluation of the policy-practice gaps. If you used another authors framework. I would state it here and define what their framework is. This will then help set up whether you are adding to their framework (theory building) or simply using their framework to understand your operational context. The latter is not really theory building but could be if the original framework of five factors doesn’t represent the constructs you found. Then you might even add a construct and develop a new framework / theory.

Methods:

I understand that you used descriptive analysis to explain who used the OFTT then interviewed 19 users about their experience. Typically, questions would be informed by the the literature. Can I please ask what are the questions you asked users? Can you provide a sample of 5 questions in the text? At the very least provide the topics you asked about and the literature they were derived from. Typically, new digital interventions will be evaluated for usability, usefulness, effectiveness, and efficacy. Did the framework you selected provide questions? If yes, please elaborate on that here. The central questions of the study line 124 are the overarching questions for the study. So, this is not confusing, I would please these at the end of the introduction, rather than in the methodology section (unless the journal prescribes where to put them).

Another major issue with the method is there is not mention of the following:

·         Did you record and transcribe the interviews or did you use a framework analysis and make notes directly from interviews?

·         How many researchers did the coding? Was it deductive or inductive?

·         What was the percentage of agreement between researchers? How was this resolved?

·         Were there other ways you addressed rigor?

Results:

A well-crafted section. One improvement would be to include a table of basic descriptive stats of the original OFTT users. Table 1 Key Informants is simply a repeat of the information you have previously provided. Either improve the table with more demographic information (eg. sex, age, medical condition, medical speciality, OFTT expertise etc) or remove it.

Discussion:

Good reference back to other literature.

Conclusion:

A well-crafted section. Can you recommend future research?

Lastly did the assessment actually change practice? It would be good to see how this assessment changed or closed the policy-practice gap in this setting.

Overall:

I feel there is a bit of repetition in this paper.

The abstract, key message and introduction have identical paragraphs. Can this be improved?

There are five times you describe how the five factors pertain to this study. It is very repetitive. Can you consider moving the table into an appendix? Try to condense the results section to explain each element once then present the overarching diagram.

Thank you for the opportunity to review Identifying Policy Gaps in a COVID-19 online tool using the five-factor framework.

Author Response

(The authors gave the same response as above.)

Reviewer 3 Report

My personal opinion would be to add some information, if it exists, about other OFTT tools in South African hospitals, or other hospitals you may know of and what were the findings in implementing them. The same questions apply for Bern hospitals if there are any OFTT tools. 

It would be interesting to see the limitations and future directions for this OFTT tool the author(s) described. 

Author Response

Reviewer 3

Reviewer Comment: The central questions of the study line 124 are the overarching questions for the study. So, this is not confusing, I would please these at the end of the introduction, rather than in the methodology section (unless the journal prescribes where to put them).

*Author response

The central question for this manuscript are different from the central questions of the study hence we fitted them into the manuscript methodology section as below;

Central questions

How well do the identified themes fit into the five-factor framework?

How well does the five-factor framework explain why and how the policy practice gaps came about?

Reviewer comment: Another major issue with the method is there is not mention of the following: Did you record and transcribe the interviews or did you use a framework analysis and make notes directly from interviews?  How many researchers did the coding? Was it deductive or inductive?  What was the percentage of agreement between researchers? How was this resolved? Were there other ways you addressed rigor?

*Author response

We thank the reviewers for raising these questions. We have addressed them. See lines 161-174

Qualitative Data analysis

All audio recordings were transcribed verbatim, analysed and triangulated with results from the quantitative data. Qualitative narratives were explored for fit with the five factors of the  analytic framework.12 Two qualitative researchers analysed the transcripts independently and developed and agreed on a code book. All the concepts fitted into the five factor themes.

Measures to ensure trustworthiness of data: To ensure dependability, the data collection process and analysis were performed iteratively, continuously adjusting our interview guide to capture newly emerging themes. Two qualitative researchers kept reflexive journals and debriefed at the end of each interview throughout the data collection phase. A thick description of participants, context and data collection process has been outlined here to ensure transferability. Data was managed and analysed with the aid of MAXQDA2020.

Lines 139-154

Qualitative data collection

The qualitative interviews were conducted with purposefully selected key informants who gave their consent during the survey. Video rather than face to face interviews were held with most participants in September 2020 due to social -distancing rules. A combination of video and telephonic interviews were conducted with three participants who had technical challenges and a telephone only interview was held with one lady, aged above 65, who had no computer access. Three face to face interviews were held with three key informants: one that was a hospital health care worker, and two key informants who worked close to Bern university hospital. A semi-structured interview guide informed by the quantitative results was used (see supplementary info). This guide was adapted iteratively throughout the data collection period. Two qualitative researchers sat in each session fielding questions in turns. All interviews were conducted in German by two researchers fluent in both English and German. The interviews lasted between 45 minutes to one and a half hours. Two audio-recorders were used in each session. All participants gave individual written consent as well as oral consent to the recording at the beginning of each session. See Table 1 and 2 for summary of Key Informants.

Results: A well-crafted section. One improvement would be to include a table of basic descriptive stats of the original OFTT users. Table 1 Key Informants is simply a repeat of the information you have previously provided. Either improve the table with more demographic information (eg. sex, age, medical condition, medical speciality, OFTT expertise etc) or remove it.

*Author response

We thank the reviewers for raising this issue. A respondents table has now been included. See above.

Discussion: Good reference back to other literature.

*Author response

Thank you

Conclusion: A well-crafted section. Can you recommend future research? Lastly did the assessment actually change practice? It would be good to see how this assessment changed or closed the policy-practice gap in this setting.

*Author response

We thank the reviewers for their interest. Yes, recommendations have been made to authorities during results presentation sessions and also through publications. See links below

https://www.frontiersin.org/articles/10.3389/fpubh.2022.901125/full

https://www.frontiersin.org/articles/10.3389/fpubh.2022.902072/full

Overall: I feel there is a bit of repetition in this paper. The abstract, key message and introduction have identical paragraphs. Can this be improved? There are five times you describe how the five factors pertain to this study. It is very repetitive. Can you consider moving the table into an appendix? Try to condense the results section to explain each element once then present the overarching diagram.

*Author response

We have attended to the reviewer suggestions and minimized repetitions as far as possible. Key messages have been removed.  

My personal opinion would be to add some information, if it exists, about other OFTT tools in South African hospitals, or other hospitals you may know of and what were the findings in implementing them. The same questions apply for Bern hospitals if there are any OFTT tools. It would be interesting to see the limitations and future directions for this OFTT tool the author(s) described. 

*Author response

We thank the authors for these comments. We have published elsewhere the limits of OFTTs. Below is a list of some of our publications in this regard

  • Michel, J.; Mettler, A.; Müller, M.; Hautz, W.E.; Sauter, T.C. A Utility Framework for COVID-19 Online Forward Triage Tools: A Swiss Telehealth Case Study. Int. J. Environ. Res. Public Health 2022, 19, 5184. https://doi.org/10.3390/ ijerph19095184
  • Michel J, Rehsmann J, Mettler A, Starvaggi C, Travaglini N, Aebi C, Keitel K. Sauter TC. Public Health communication: attitudes, experiences and lessons 1 learned from users of a COVID-19 digital triage tool for children. Front. Public Health 10:902072. doi: 10.3389/fpubh.2022.902072
  • Michel J, Mettler A, Starvaggi C, Travaglini N, Aebi C, Keitel K and Sauter TC (2022) The Utility of a Pediatric COVID-19 Online Forward Triage Tool in Switzerland. Front. Public Health 10:902072. doi: 10.3389/fpubh.2022.902072
  • Michel J, Kilb T, Mettler A, Müller M, Hautz WE, Hautz SC and Sauter TC (2022) The Utility of an Online Forward Triage Tool During the SARS-CoV-2 Pandemic: Health Care Provider and Health Authority Perspectives. Front. Public Health 10:845996. doi: 10.3389/fpubh.2022.845996
  • Michel J, Mettler A, Stuber R, Müller M, Ricklin ME, Jent P, Hautz WE, Sauter TC. Effects and utility of an online forward triage tool during the SARS-CoV-2 pandemic: a mixed method study and patient perspectives, Switzerland. BMJ Open 2022;0: e059765. doi:10.1136/ bmjopen-2021-059765
  • Janet Michel, Wolf E Hautz, Thomas C Sauter. Telemedicine and online platforms as an opportunity to optimize qualitative data collection, explore and understand disease pathways in a novel pandemic like COVID-19. Journal of the International Society for Telemedicine and Ehealth 2020, 8, e9 (1-4).
  • Janet Michel, Annette Mettler, Wolf Hautz, Thomas Sauter. What is the optimal length of an Online Forward Triage Tool? The need for a framework. J Glob Health 2020 Dec;10(2):0203103.
  • Michel J, Stuber R, Müller M, et al. COVID-19 and HIV testing: different viruses but similar prejudices and psychosocial impacts. Journal of Global Health Reports. 2021;5: e2021022.

Reviewer 3

Reviewer Comment: The central questions of the study line 124 are the overarching questions for the study. So, this is not confusing, I would please these at the end of the introduction, rather than in the methodology section (unless the journal prescribes where to put them).

*Author response

The central question for this manuscript are different from the central questions of the study hence we fitted them into the manuscript methodology section as below;

Central questions

How well do the identified themes fit into the five-factor framework?

How well does the five-factor framework explain why and how the policy practice gaps came about?

Reviewer comment: Another major issue with the method is there is not mention of the following: Did you record and transcribe the interviews or did you use a framework analysis and make notes directly from interviews?  How many researchers did the coding? Was it deductive or inductive?  What was the percentage of agreement between researchers? How was this resolved? Were there other ways you addressed rigor?

*Author response

We thank the reviewers for raising these questions. We have addressed them. See lines 161-174

Qualitative Data analysis

All audio recordings were transcribed verbatim, analysed and triangulated with results from the quantitative data. Qualitative narratives were explored for fit with the five factors of the  analytic framework.12 Two qualitative researchers analysed the transcripts independently and developed and agreed on a code book. All the concepts fitted into the five factor themes.

Measures to ensure trustworthiness of data: To ensure dependability, the data collection process and analysis were performed iteratively, continuously adjusting our interview guide to capture newly emerging themes. Two qualitative researchers kept reflexive journals and debriefed at the end of each interview throughout the data collection phase. A thick description of participants, context and data collection process has been outlined here to ensure transferability. Data was managed and analysed with the aid of MAXQDA2020.

Lines 139-154

Qualitative data collection

The qualitative interviews were conducted with purposefully selected key informants who gave their consent during the survey. Video rather than face to face interviews were held with most participants in September 2020 due to social -distancing rules. A combination of video and telephonic interviews were conducted with three participants who had technical challenges and a telephone only interview was held with one lady, aged above 65, who had no computer access. Three face to face interviews were held with three key informants: one that was a hospital health care worker, and two key informants who worked close to Bern university hospital. A semi-structured interview guide informed by the quantitative results was used (see supplementary info). This guide was adapted iteratively throughout the data collection period. Two qualitative researchers sat in each session fielding questions in turns. All interviews were conducted in German by two researchers fluent in both English and German. The interviews lasted between 45 minutes to one and a half hours. Two audio-recorders were used in each session. All participants gave individual written consent as well as oral consent to the recording at the beginning of each session. See Table 1 and 2 for summary of Key Informants.

Results: A well-crafted section. One improvement would be to include a table of basic descriptive stats of the original OFTT users. Table 1 Key Informants is simply a repeat of the information you have previously provided. Either improve the table with more demographic information (eg. sex, age, medical condition, medical speciality, OFTT expertise etc) or remove it.

*Author response

We thank the reviewers for raising this issue. A respondents table has now been included. See above.

Discussion: Good reference back to other literature.

*Author response

Thank you

Conclusion: A well-crafted section. Can you recommend future research? Lastly did the assessment actually change practice? It would be good to see how this assessment changed or closed the policy-practice gap in this setting.

*Author response

We thank the reviewers for their interest. Yes, recommendations have been made to authorities during results presentation sessions and also through publications. See links below

https://www.frontiersin.org/articles/10.3389/fpubh.2022.901125/full

https://www.frontiersin.org/articles/10.3389/fpubh.2022.902072/full

Overall: I feel there is a bit of repetition in this paper. The abstract, key message and introduction have identical paragraphs. Can this be improved? There are five times you describe how the five factors pertain to this study. It is very repetitive. Can you consider moving the table into an appendix? Try to condense the results section to explain each element once then present the overarching diagram.

*Author response

We have attended to the reviewer suggestions and minimized repetitions as far as possible. Key messages have been removed.  

My personal opinion would be to add some information, if it exists, about other OFTT tools in South African hospitals, or other hospitals you may know of and what were the findings in implementing them. The same questions apply for Bern hospitals if there are any OFTT tools. It would be interesting to see the limitations and future directions for this OFTT tool the author(s) described. 

*Author response

We thank the authors for these comments. We have published elsewhere the limits of OFTTs. Below is a list of some of our publications in this regard

  • Michel, J.; Mettler, A.; Müller, M.; Hautz, W.E.; Sauter, T.C. A Utility Framework for COVID-19 Online Forward Triage Tools: A Swiss Telehealth Case Study. Int. J. Environ. Res. Public Health 2022, 19, 5184. https://doi.org/10.3390/ ijerph19095184
  • Michel J, Rehsmann J, Mettler A, Starvaggi C, Travaglini N, Aebi C, Keitel K. Sauter TC. Public Health communication: attitudes, experiences and lessons 1 learned from users of a COVID-19 digital triage tool for children. Front. Public Health 10:902072. doi: 10.3389/fpubh.2022.902072
  • Michel J, Mettler A, Starvaggi C, Travaglini N, Aebi C, Keitel K and Sauter TC (2022) The Utility of a Pediatric COVID-19 Online Forward Triage Tool in Switzerland. Front. Public Health 10:902072. doi: 10.3389/fpubh.2022.902072
  • Michel J, Kilb T, Mettler A, Müller M, Hautz WE, Hautz SC and Sauter TC (2022) The Utility of an Online Forward Triage Tool During the SARS-CoV-2 Pandemic: Health Care Provider and Health Authority Perspectives. Front. Public Health 10:845996. doi: 10.3389/fpubh.2022.845996
  • Michel J, Mettler A, Stuber R, Müller M, Ricklin ME, Jent P, Hautz WE, Sauter TC. Effects and utility of an online forward triage tool during the SARS-CoV-2 pandemic: a mixed method study and patient perspectives, Switzerland. BMJ Open 2022;0: e059765. doi:10.1136/ bmjopen-2021-059765
  • Janet Michel, Wolf E Hautz, Thomas C Sauter. Telemedicine and online platforms as an opportunity to optimize qualitative data collection, explore and understand disease pathways in a novel pandemic like COVID-19. Journal of the International Society for Telemedicine and Ehealth 2020, 8, e9 (1-4).
  • Janet Michel, Annette Mettler, Wolf Hautz, Thomas Sauter. What is the optimal length of an Online Forward Triage Tool? The need for a framework. J Glob Health 2020 Dec;10(2):0203103.
  • Michel J, Stuber R, Müller M, et al. COVID-19 and HIV testing: different viruses but similar prejudices and psychosocial impacts. Journal of Global Health Reports. 2021;5: e2021022.

Round 2

Reviewer 1 Report

The manuscript has been modified in accordance with previous recommendations. The methodology of the manuscript is explained in more detail, so the concept of the study itself is clearly defined. All implemented changes contribute to the improvement of coherence and scientific significance.